# Liraglutide Attenuates Glucolipotoxicity-Induced RSC96 Schwann Cells’ Inflammation and Dysfunction

**DOI:** 10.3390/biom12101338

**Published:** 2022-09-21

**Authors:** Edy Kornelius, Sing-Hua Tsou, Ching-Chi Chang, Ying-Jui Ho, Sheng-Chieh Lin, Wei-Liang Chen, Chien-Ning Huang, Chih-Li Lin

**Affiliations:** 1Department of Internal Medicine, Division of Endocrinology and Metabolism, Chung Shan Medical University Hospital, Taichung 402367, Taiwan; 2School of Medicine, Chung Shan Medical University, Taichung 402367, Taiwan; 3Department of Medical Research, Chung Shan Medical University Hospital, Taichung 402367, Taiwan; 4Department of Psychiatry, Chung Shan Medical University Hospital, Taichung 402367, Taiwan; 5School of Psychology, Chung Shan Medical University, Taichung 402367, Taiwan; 6Department of Orthopedics, Chung Shan Medical University Hospital, Taichung 402367, Taiwan; 7Institute of Medicine, Chung Shan Medical University, Taichung 402367, Taiwan; 8Department of Internal Medicine, Division of Gastroenterology and Hepatology, Chung Shan Medical University Hospital, Taichung 402367, Taiwan

**Keywords:** diabetic neuropathy, GLP-1, insulin signaling, Liraglutide, Schwann cell

## Abstract

Diabetic neuropathy (DN) is a type of sensory nerve damage that can occur in patients with diabetes. Although the understanding of pathophysiology is incomplete, DN is often associated with structural and functional alterations of the affected neurons. Among all possible causes of nerve damage, Schwann cells (SCs) are thought to play a key role in repairing peripheral nerve injury, suggesting that functional deficits occurring in SCs may potentially exhibit their pathogenic roles in DN. Therefore, elucidating the mechanisms that underlie this pathology can be used to develop novel therapeutic targets. In this regard, glucagon-like peptide-1 receptor agonists (GLP-1 RAs) have recently attracted great attention in ameliorating SCs’ dysfunction. However, the detailed mechanisms remain uncertain. In the present study, we investigated how GLP-1 RA Liraglutide protects against RSC96 SCs dysfunction through a diabetic condition mimicked by high glucose and high free fatty acid (FFA). Our results showed that high glucose and high FFAs reduced the viability of RSC96 SCs by up to 51%, whereas Liraglutide reduced oxidative stress by upregulating antioxidant enzymes, and thus protected cells from apoptosis. Liraglutide also inhibited NFκB-mediated inflammation, inducing SCs to switch from pro-inflammatory cytokine production to anti-inflammatory cytokine production. Moreover, Liraglutide upregulated the production of neurotrophic factors and myelination-related proteins, and these protective effects appear to be synergistically linked to insulin signaling. Taken together, our findings demonstrate that Liraglutide ameliorates diabetes-related SC dysfunction through the above-mentioned mechanisms, and suggest that modulating GLP-1 signaling in SCs may be a promising strategy against DN.

## 1. Introduction

Diabetic neuropathy (DN) is a type of nerve damage that occurs in people with diabetes. It is estimated that about half of people with diabetes will eventually develop neuropathy. DN is primarily a sensory neurological disorder in which patients typically present with abnormal sensations of the skin, such as numbness or tingling (paresthesia). Some neuropathic pain symptoms that are particularly common in DN include hypersensitive reaction to touch (allodynia), increased sensitivity to noxious stimuli (hyperalgesia), and spontaneous pain. These symptoms often lead to depression, anxiety, and sleep disorders, which can significantly reduce quality of life [1]. Although the pathophysiology of DN remains incomplete, it is known to be primarily related to the structural and functional abnormalities of the sensory neuron axons in the peripheral nervous system (PNS) [2]. There are several forms of PNS damage, and the most common type is diabetic distal symmetric polyneuropathy, which accounts for approximately 75% of all DN cases [3]. Therefore, it is generally believed that the progression of DN can be managed by preventing further PNS neuronal damage. In this regard, Schwann cells (SCs) in the PNS are recognized to play an essential role in the survival and functions of neurons [4]. SCs are the principal glia of the PNS that form a multilayer myelin sheath wrapping around the axon of neurons. The myelin sheath increases electrical impulses, leading to effective transmission and protecting the axons of neurons. Therefore, it is clear that SCs play a key role in maintaining neural structure and function, suggesting that the defective damage responses of SCs may be the potential mechanism for regulating the pathogenesis of DN [5].

Studies have shown that, in the progression of DN, glucolipotoxicity is one of the main causes of continued deterioration [6]. Chronic glucolipotoxicity in T2D leads to the induction of oxidative stress and inflammation, which may contribute to cellular dysfunction and apoptosis [7]. In particular, SCs protect against oxidative stress and modulate immune responses by releasing various cytokines, and thus display neuroprotective effects [8]. This supports the hypothesis that defects in SC responses to injury may contribute to the pathogenesis of DN. In fact, it has been reported that impaired insulin-PI3K-Akt signaling in SCs significantly inhibits myelination and contributes to DR [9]. This indicates that the restoration of the insulin-signaling capacities of SCs may help to improve myelination and other normal physiological functions, and thus may be beneficial for alleviating DR [10]. Therefore, agents with attenuation of the insulin signaling blockade are also considered, showing the effect of ameliorating DN-induced SCs dysfunction and apoptosis. In this regard, glucagon-like peptide-1 receptor agonists (GLP-1 RAs) have recently attracted great attention. GLP-1 is an incretin hormone that exhibits insulinotropic effects [11]. In particular, the GLP-1 receptors are abundantly expressed in SCs, implying a critical role for GLP-1 signaling in the regulation of physiological functions [12]. In fact, activation of GLP-1 signaling via Exendin-4 has recently been found to promote the survival and myelination of SCs, and may therefore be relevant for alleviating DN [13]. However, the detailed anti-DN mechanisms of the GLP-1 analog are still not fully characterized. In the present study, we sought to determine whether GLP-1 RA Liraglutide exerts a protective effect on SCs. Liraglutide is a homologous analog with the highest similarity (~97%) to human GLP-1, acting as an ideal agonist at the GLP-1 receptor. By using an in vitro model of glucolipotoxicity, we found that the activation of GLP-1 signaling in rat RSC96 SCs reduced glucolipotoxicity-induced inflammation and oxidative stress, and increased the ability of SCs to produce neurotrophic factors synergistically with insulin signaling.

## 2. Materials and Methods

### 2.1. Materials

Chemicals such as 3-(4,5-dimethylthiazol-2-yl)-2,5-diphenyltetrazolium bromide (MTT), 4′,6-diamidino-2-phenylindole (DAPI), 2′,7′dichlorofluorescin diacetate (DCFH-DA), LY294002, and JC-1 were purchased from Sigma (München, Germany). Antibodies against β-actin (sc-47778), Akt (sc-8312), p-Akt (sc-33437), caspase 1 (sc-56036), poly(ADP-ribose) polymerase (PARP) (sc-7150), p65 (sc-8008), p-p65 (sc-136548), IκBα (sc-1643), AMPK (sc-74461), p-AMPK (sc-33524), RAGE (sc-365154), and 4-hydroxynonenal (4-HNE) (sc-130083) were obtained from Santa Cruz Biotechnology (Santa Cruz, CA, USA). Antibodies against SOD1 (#2770S), SOD2 (GTX116093), catalase (GTX110704), and Sirt1 (GTX61042) were purchased from GeneTex (Irvine, CA, USA). Antibodies against NLRP3 (NBP2-12446), ASC (NBP1-89656), PLP (NBP1-87781), MBP (NB600-717), and MPZ (NB100-1607) were obtained from Novus Biologicals (Littleton, CO, USA). Antibodies against cleaved-caspase 3 (#9661), IRS-1 (194320), and p-IRS-1 (05-1087) were acquired from Cell Signaling Technology (Danvers, MA, USA). Primary antibodies were used at 1:200 (ICC) or 1:1000 (WB) dilutions in 0.1% Tween 20 blocking solution, and secondary antibodies were used at 1:1000–1:5000 dilutions, depending on the condition of the individual samples.

### 2.2. Cell Culture and Viability Assay

Rat RSC96 Schwann cells were purchased from the Bioresource Collection and Research Center, Food Industry Research and Development Institute (Hsinchu, Taiwan). Cells were cultured in Dulbecco’s Modified Eagle Medium (DMEM; Gibco), supplemented with 10% fetal calf serum, antibiotics (100 units/mL penicillin, 100 µg/mL streptomycin), and 2 mM L-glutamine, and kept in humidified air containing 5% CO_2_ at 37 °C. For the viability assay, cells were treated with tetrazolium salt MTT for 30 min, and then analyzed spectrophotometrically at 550 nm. Viability was determined as a percent of control cells treated with the vehicle alone.

### 2.3. Western Blot Analysis

Samples were prepared using Gold Lysis Buffer (50 mM Tris-HCl, pH 8.0, 5 mM ethylenediaminetetraacetic acid, 150 mM NaCl, 0.5% Nonidet P-40, 0.5 mM phenylmethylsulfonyl fluoride, and 0.5 mM dithiothreitol) to obtain whole cell extracts. After quantification of the protein concentration within each sample, equal amounts of protein in whole cell lysates were separated by 8–12% sodium dodecyl sulfate (SDS)-polyacrylamide gel electrophoresis, and then transferred to a polyvinylidene difluoride (PVDF) membrane (Millipore). After blocking, the membranes were sequentially probed with primary antibodies, followed by secondary antibodies conjugated to horseradish peroxidase. Finally, protein signals on the PVDF membrane were shown using Amersham ECL detection reagents, and the images were acquired using the AI600 Imaging System (GE Healthcare, Chicago, IL, USA).

### 2.4. Measurement of Reactive Oxygen Species (ROS)

To measure intracellular ROS levels, cells were treated with 20 μM of DCFH-DA for 0.5 h at 37 °C under 5% CO_2_. After incubation, cells were harvested and immediately washed twice with PBS to remove background signals. For the quantification of intracellular ROS content, we conducted our analysis at the indicated times with a multi-detection reader at excitation and emission wavelengths of 485 and 535 nm (SpectraMax iD5 microplate reader, Molecular Devices, Sunnyvale, CA, USA). The relative fluorescence intensity was taken as the average of values from three repeated experiments. The fluorescent images were also collected by fluorescence microscopy (DP72/CKX41, Olympus), and all images used the same fluorescent conditions and exposure time.

### 2.5. Analysis of Mitochondrial Membrane Potential by JC-1 Staining

To assess mitochondrial membrane potential, we incubated the cells with a fresh medium containing 1 µM JC-1 for 30 min at 37 °C. At the end of the incubation, the cells were washed to remove the staining medium, and then imaged using an inverted fluorescence microscope (DP72/CKX41, Olympus). Image Pro Plus 6.0 (Media Cybernetics, Rockville, MD, USA) software was used to measure red and green fluorescence, and the results were expressed as the ratio of the mean red/green fluorescence intensities. All results are calculated using 5 random non-adjacent images per group for statistical analysis.

### 2.6. Immunofluorescence Assay

To analyze the intracellular distribution of NFκB, the cells were harvested and fixed with ice-cold methanol with 4% paraformaldehyde for 24 h. After three washes with PBS, the cells were permeabilized for 10 min amd blocked with blocking solution for 2 h; then, they were incubated with blocking solution for 2 h, incubated with the anti-NFκB p65 antibody (1:200) overnight, and then washed with PBS. After incubation with the secondary antibody for 1 h, the cells were stained with DAPI solution (1 ng/mL in Mcllvaine’s buffer, pH 7.0) for 15 min at RT. Fluorescent images were acquired using a fluorescent microscope (DP80/BX53, Olympus, Tokyo, Japan).

### 2.7. mRNA Expression Analysis by Reverse-Transcription Quantitative PCR (qPCR)

Total RNA was extracted from cells using the RNeasy Kit (Qiagen, Germantown, MD, USA) according to the manufacturer protocol. After purification of the RNA, mRNA was reverse transcribed into cDNA using the TProfessional Thermocycler Biometra (Göttingen, Germany) according to the manufacturer’s recommendations. A qPCR was then performed on an ABI 7300 Sequence Detection System (Applied Biosystems, Foster City, CA, USA) using Power SYBR Green PCR Master Mix (Applied Biosystems). The reverse transcription process was performed using the following temperature parameters: initial denaturation at 95 °C for 10 min, 40 cycles of denaturation at 95 °C for 15 s, annealing at 60 °C for 1 min, and a dissociation phase at 95 °C for 15 s, 60 °C for 15 s, and 95 °C for 15 s. The primer pairs are shown in Table 1. The mRNA levels were normalized to glyceraldehyde 3-phosphate dehydrogenase (GAPDH). Each cDNA sample was tested in triplicate. Values of relative mRNA expression were obtained using Sequence Detection Systems software (Sequence Detection Systems 1.2.3 and 7300 Real-Time PCR System; Applied Biosystems) with a cycle threshold (delta-delta Ct) method.

### 2.8. Statistical Analysis

All data are presented as the mean ± standard error of the mean (±SEM). Data were statistically analyzed using analysis of variance followed by Dunnett’s post hoc test for multiple comparisons using SPSS v25.0 statistical software (SPSS, Inc., Chicago, IL, USA). Differences were considered statistically significant using the enclosed * for *p* < 0.05 and ** for *p* < 0.01.

## 3. Results

### 3.1. GLP-1 RA Liraglutide Attenuates Glucolipotoxicity-Induced Apoptosis in RSC96 Schwann Cells

It is now apparent that increased levels of glucose and free fatty acids (FFAs) are associated with the development of diabetic complications, including DN [14]. To explore the possible effects of glucolipotoxicity on SCs, we first analyzed the changes in RSC96 SCs’ viability caused by different concentrations of glucose and FFA. In order to simulate the composition of FFAs in the human body, we mixed palmitic acid (the most abundant saturated FA) and oleic acid (the most abundant unsaturated FA) in a ratio of 1:2, and conjugated with bovine serum albumin (BSA) to increase its solubility in the medium [15]. As shown in Figure 1, the results of the MTT assays demonstrated that, as the concentration of FFAs increases, the cell viability gradually decreases, and an additional 30 mM of glucose reduces the cell viability significantly further. In addition, no significant difference in cell viability was observed between sucrose-treated and control cells, indicating that the hypertonic environment can be excluded as the main cause of cytotoxicity. Based on this result, we used 30 mM glucose and 250 mM FFAs as the condition to induce glucolipotoxicity in all subsequent experiments. Next, we confirmed by Western blotting that RSC96 SCs can express the GLP-1 receptor, and its expression was not affected by high glucose or/and high FFAs (Figure 1b). Although previous studies have suggested that activation of GLP-1 signaling may be beneficial for alleviating DN, the effect of Liraglutide on SCs is still unclear. As a result, we investigated whether Liraglutide shows a protective effect on SCs under glucolipotoxicity. As shown in Figure 1c, the results of MTT showed that Liraglutide alone displays no significant effect on SCs. However, Liraglutide significantly protected SCs from glucolipotoxicity at a concentration of 0.01 to 0.1 μM. Similarly, the results of Western blotting also demonstrated that Liraglutide effectively reduces the activation of two proapoptotic markers, PARP and caspase 3. These findings support the protective role of GLP-1 signaling in attenuating glucolipotoxicity-induced apoptosis for SCs.

### 3.2. Liraglutide Decreases Oxidative Stress by Upregulating Antioxidant Defense Mechanisms

Although multiple causes can be involved in the pathogenesis of DN, previous studies have found that glucolipotoxicity-induced oxidative stress contributes directly and indirectly to SC dysfunction [16,17]. To investigate whether Liraglutide affects changes in oxidative stress levels in RSC96 SCs under glucolipotoxicity, we used the DCFH-DA florescent probe to detect ROS production by flow cytometry. As shown in Figure 2a, intracellular ROS levels increased to their highest after 12 h of treatment with high glucose and FFAs. In contrast, co-treatment with Liraglutide markedly inhibited intracellular ROS accumulation caused by glucolipotoxicity. The results of the fluorescence microscopic images also confirmed that Liraglutide potently inhibits glucolipotoxicity-induced intracellular ROS levels (Figure 2b). Next, we attempted to clarify the possible mechanism in reducing ROS. The results of the Western blot, presented in Figure 2c, show that high glucose and FFAs markedly inhibited AMPK pThr^172^ phosphorylation and Sirt1 protein expression, while Liraglutide could effectively restore the above conditions. This result is similar to our previous finding [15] that, when Liraglutide restores the AMPK-Sirt1 pathway, it upregulates downstream antioxidant defense enzymes including SOD1/2 and catalase, thereby reducing glucolipotoxicity-induced ROS levels. Moreover, the results of JC-1 staining also suggested that the antioxidant capacity restored by Liraglutide can reduce the glucolipotoxicity-impaired mitochondrial membrane potential (Figure 2d). The above results indicate that Liraglutide may indeed attenuate glucolipotoxicity-induced oxidative stress by restoring intracellular antioxidant defense mechanisms.

### 3.3. Liraglutide Downregulates NFκB-Mediated Inflammation under Glucolipotoxicity

It is known that SC-mediated inflammatory responses play an important role in nerve damage in DN [18]. Therefore, we investigated changes in NFκB transcriptional regulation in the inflammatory response. As shown in Figure 3a, the results of Western blotting demonstrated that high glucose and FFAs stimulate IκBα degradation and phosphorylate the NFκB p65 subunit at Ser^536^, suggesting that glucolipotoxicity promotes the activation of NFκB. In contrast, 0.1 μM Liraglutide effectively inhibited the activation of NFκB under glucolipotoxicity, indicating that Liraglutide can effectively suppress the inflammatory response of SCs. The results of immunofluorescence staining also confirmed that Liraglutide inhibits the entry of the NFκB p65 subunit into the nucleus under glucolipotoxicity (Figure 3b), suggesting that NFκB-mediated inflammation may indeed be attenuated. In order to further clarify whether the inflammatory response is inhibited by Liraglutide, we selected some markers downstream of NFκB for analysis. As shown in Figure 3c, the results of the qPCR revealed that some SC-associated pro-inflammatory cytokines, including IL-1β, TNFα and monocyte chemoattractant protein-1 (MCP-1), were significantly increased under high glucose and FFAs. However, co-treatment with Liraglutide effectively reduced the mRNA expression of these pro-inflammatory cytokines. Interestingly, Liraglutide also restored the mRNA expression levels of anti-inflammatory cytokines such as IL-4, IL-10, and IGF-1 that had been suppressed by glucolipotoxicity, showing that the activation of GLP-1 signaling may mediate a shift in proinflammatory SCs towards an anti-inflammatory phenotype. In addition, excessive ROS in diabetes have been found to trigger NOD-like receptor protein 3 (NLRP3) inflammasome activation in SCs [19]. In order to clarify the underlying mechanism by which the inflammation is inhibited by Liraglutide, we analyzed the protein markers associated with inflammasome activation. As shown in Figure 3d, the Western blotting results demonstrated that high glucose and FFAs increase inflammasome components, including NLRP3, apoptosis-associated speck-like protein containing a caspase recruitment domain (ASC), and cleaved-caspase-1. In contrast, co-treatment with Liraglutide markedly suppressed the levels above key components of the inflammasome, providing substantial evidence that GLP-1 signaling plays a key inhibitory role in glucolipotoxicity-induced SC inflammation.

### 3.4. Liraglutide Enhances the Physiological Function of SCs through Insulin-AKT Signaling

There is evidence that the dysfunction of SCs plays an important role in the development of DN, resulting in changes in the microenvironment and thus leading to neuronal damage. Specifically, disrupting insulin signaling is one of the most important features of SC dysfunction [9]; as such, we analyzed the phosphorylation changes in Akt, a key kinase downstream of insulin receptors. As shown in Figure 4a, immunoblotting results demonstrated that high levels of glucose and FFAs markedly reduced the phosphorylation of Akt at Ser^473^, indicating that insulin signaling was indeed significantly inhibited. However, adding Liraglutide effectively restored the phosphorylation of Akt; this improvement is even more significant if insulin is added at the same time, implying that Liraglutide may exert its protective effect synergistically with insulin signaling. The results of MTT also showed that the additional insulin markedly enhanced the protective efficacy of Liraglutide on RSC96 SCs. In particular, the PI3K-specific inhibitor LY294002 significantly reduced the effect of Liraglutide, once again verifying that insulin signaling may be involved in the protective mechanisms of SCs against glucolipotoxicity (Figure 4b). It is known that SCs can provide a growth-supportive microenvironment by secreting neurotrophic factors to maintain neuronal function and survival. To assess the ability of SCs to support peripheral neurons, we analyzed the mRNA synthesis of neurotrophic factors. The results of the qPCR shown in Figure 4c reveal that, for some known SC-secreted neurotrophic factors, including the ciliary neurotrophic factor (CNTF), the nerve growth factor (NGF), neurotrophin-3 (NT-3), and the brain-derived neurotrophic factor (BDNF), the mRNA expression levels were significantly decreased under glucolipotoxicity. However, Liraglutide and insulin could also effectively restore the mRNA expression of these neurotrophic factors. Finally, we measured the changes in protein expression associated with myelination, the main physiological function of SCs. As shown in Figure 4d, some essential myelin components, including the proteolipid protein (PLP), the myelin basic protein (MBP), and myelin protein zero (MPZ), were markedly reduced, while inflammation-associated demyelination markers for the receptor for advanced glycation endproducts (RAGE) were obviously elevated under glucolipotoxicity. In contrast, Liraglutide and insulin effectively restored the expression levels of the various above-mentioned proteins to a state close to normal. These findings show that Liraglutide displays the effect of improving SC dysfunction induced by high glucose and FFAs. Moreover, LY294002 counteracted the protective effect of Liraglutide, which also implies that the action mechanism may function dependently through the insulin signaling pathway.

## 4. Discussion

In most cases, insulin signaling dysregulation may be a triggering or exacerbating factor for neuropathic changes in sensory neurons. This is especially important because sensory nerve damage is a major pathogenic process in DN [20]. Both hyperglycemia and hyperlipidemia are common symptoms in diabetic patients, and the above symptoms often cause obvious insulin resistance to peripheral tissues [21]. However, there is still no definitive answer to the question of how a loss of normal insulin action impacts glial cells. In the peripheral nervous system, SCs are the major glial cell type, and play essential roles in the formation, maintenance, and regeneration of the myelin sheath. All of the above SC-related functions play a key role in DN; therefore, when SC is damaged and leads to dysfunction, it may be a major contributor to the acceleration of disease progression. In fact, studies have indicated that the insulin signaling blockade may also be one of the triggers for SCs dysfunction. For example, impaired SC metabolism due to insulin resistance underlies myelin defects, which deprives neurons of protection and contributes to DN [9]. Based on this premise, increased insulin signaling in SCs may be beneficial in attempts to delay or even reverse DN pathogenesis. In this regard, some anti-diabetic drugs, particularly compounds associated with the amelioration of the insulin signaling blockade, are also thought to display the effect of improving DN-induced SC defects. As a matter of fact, Takaku et al. recently found that GLP-1RA exendin-4 promotes the survival and myelination of SCs by enhancing PI3K/Akt signaling, so it may have the effect of the prevention and restoration of DPN and other PNS lesions [13]. In our present study, we further revealed that the activation of GLP-1 signaling by Liraglutide reduces glucolipotoxicity-induced ROS by increasing the AMPK-Sirt1 antioxidant enzymes, and also suppresses the NFκB-mediated inflammatory response of SCs. In addition, Liraglutide also enhanced the abilities of SCs in terms of the production of neurotrophic factors and the synthesis of essential myelin components under glucolipotoxicity, and these effects are dependent on the activity of insulin-Akt signaling. Moreover, it is already known that excessive ROS accumulation and inflammatory responses in SCs are some of the main causes of diabetic sensory neuron damage [22]. Consistent with these findings, our results provided an in-depth understanding of how GLP-1 RA Liraglutide effectively attenuates glucolipotoxicity-induced ROS accumulation and inflammation by enhancing insulin signaling, suggesting that the activation of GLP-1 signaling in SCs could indeed be a possible strategy to mitigate DN syndrome.

It is known that, in addition to the formation of myelin sheath, SCs can also play a crucial role in maintaining neuronal function by secreting a variety of extracellular regulators. Under stress, SCs secrete these substances, most of which belong to neurotrophic factors in reducing the damage of neurons [23]. Once this neuron-protective mechanism is defective, the nerve damage may be further aggravated. In previous studies, it was found that the ability of SCs to secrete the neurotrophic factors NGF and NT-3 is significantly decreased in murine models of both type 1 and type 2 diabetes [24]. In line with this, metabolic disturbances, such as diabetes-induced glucolipotoxicity, have been found to interfere with SCs and lead to the subsequent depletion of neurotrophic factor production [4]. Accordingly, our results support the idea that the ability of SCs to synthesize neurotrophic factors is indeed responsive to diabetes-induced neuronal stress, and thus may be involved in DN. Interestingly, our results also demonstrated that the ability of SCs to produce neurotrophic factors seems to be highly correlated with their insulin signaling sensitivity. That is, when insulin resistance occurs in SCs, the expression of neurotrophic factors including CTNF, NGF, NT-3, BDNF is significantly suppressed, and the expression of myelination proteins is also decreased. Although the detailed role of insulin signaling in SCs is still unclear, our study confirmed that impaired insulin signaling in SCs may be one of the main reasons for its functional impairment; this is similar to the results of other studies [9]. In addition, many reports have also shown that GLP-1 signaling displays a positive effect in terms of increasing insulin sensitivity [25], which can also help to explain the possible mechanism whereby Liraglutide promotes the protection or effect of SCs, and thus provides a possible strategy for increasing the efficacy of future treatments of DN.

Another important finding of this study is that Liraglutide can inhibit the inflammatory response of SCs induced by glucolipotoxicity. Different mechanisms are known to be involved in DN, the most critical of which are oxidative stress, inflammation, and mitochondrial dysfunction [26]. SCs are thought to play a central role in the overall response of T2D inflammation, and some studies have even found that excessive inflammation can cause SC apoptosis, thereby increasing demyelination and accelerating the progression of DN [18]. Although not all patients with DN will develop a form of demyelinating neuropathy, severely affected patients have been shown to develop inflammatory demyelination, especially if the persistence of the inflammatory response significantly increases the likelihood of adverse effects such as neuropathic pain [27]. In contrast, in addition to the traditional notion that GLP-1 signaling primarily regulates the blood glucose balance, many recent studies have also revealed its efficacy in alleviating chronic inflammation [28]. Furthermore, the activation of GLP-1 signaling in SCs has also been shown to improve their cellular function, resulting in neuroprotective effects that reduce the severity of DN [29]. In the case of T2D, SCs initiate and regulate local immune responses by secreting cellular inflammatory factors and chemokines, which upregulates the immune response and thus leads to inflammatory neuropathy [30]. As a result, maintaining the balance of SCs by adjusting GLP-1 signaling, without tending to an excessive inflammatory response, could be a reasonable therapeutic strategy for DN treatment. In fact, some GLP-1 RAs have been suggested as promising myelination-inducible and anti-demyelinating agents for SCs [25]. Taken together, our findings reveal for the first time how T2D-related glucolipotoxicity induces dysfunction in SCs, and also demonstrate the remarkable efficacy of GLP-1 RA Liraglutide in increasing antioxidant defense capacity, reducing NFκB-mediated inflammation, and improving physiological function. Since current treatments for peripheral nerve damage due to DN are generally ineffective, based on the current findings, further research on GLP-1 and the insulin signaling of SCs should be conducted. Such research will deepen our understanding of SC biology, and thus help us to identify novel and effective DN treatment strategies.

## Figures and Tables

**Figure 1 biomolecules-12-01338-f001:**
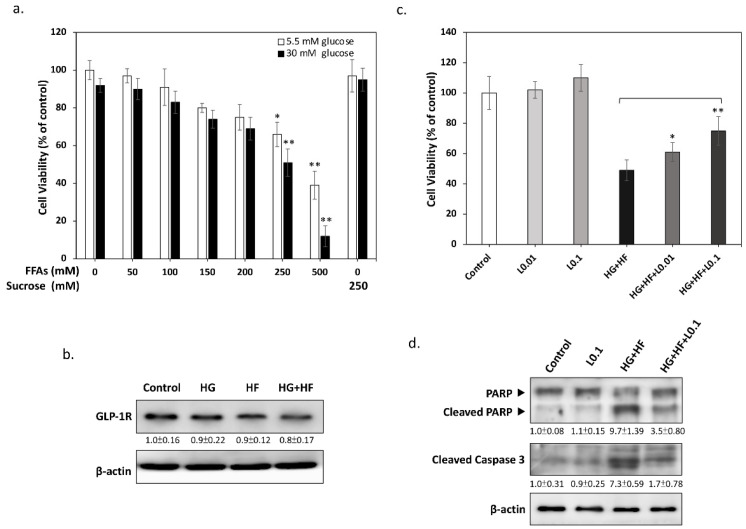
GLP-1 RA Liraglutide attenuates the glucolipotoxicity-induced apoptosis of RSC96 Schwann cells. (**a**) Cell viability was measured by an MTT assay after 24 h of treatment in the indicated conditions. The results showed that cell viability decreased significantly with increasing FFA concentrations and was more pronounced after the addition of 30 mM glucose. (**b**) Western blot analysis showed that RSC96 SCs express the GLP-1 receptor and are unaffected by glucolipotoxicity. (**c**) Liraglutide at concentrations of 0.01 to 0.1 μM significantly enhanced the survival of SCs under glucolipotoxicity. (**d**) Western blot analysis demonstrated that Liraglutide inhibits the cleavage of caspase 3 and PARP, two apoptotic markers known to be involved in glucolipotoxicity. All values are presented as the mean ± SEM. Significant difference was determined by using the multiple comparisons of Dunnett’s post hoc test for * *p* < 0.05 and ** *p* < 0.01.

**Figure 2 biomolecules-12-01338-f002:**
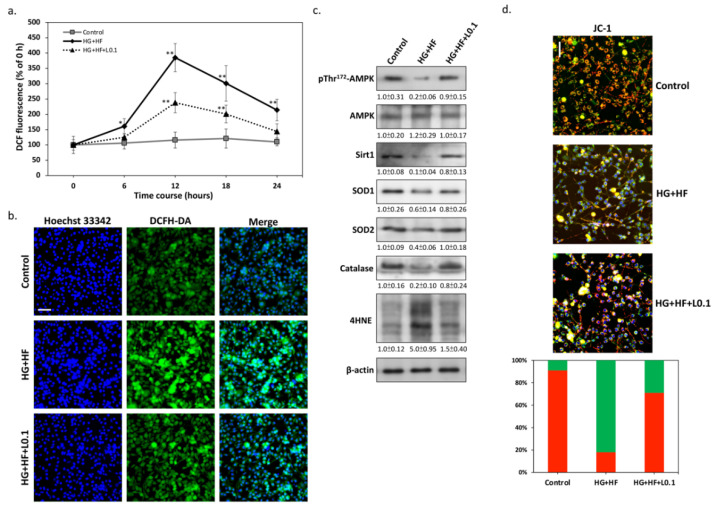
Liraglutide reduces oxidative stress by upregulating antioxidant enzymes through the AMPK-Sirt1 pathway. (**a**) Using DCFH-DA as a fluorescent indicator, quantitative data by flow cytometry showed that Liraglutide significantly inhibited ROS accumulation in RSC96 cells under glucolipotoxicity from 12 h to 24 h. (**b**) Representative images of intracellular ROS levels after DCFH-DA staining at 12 h. Scale bar: 100 μm. (**c**) The results of Western blotting revealed that Liraglutide stimulates AMPK-Sirt1 signaling, thereby restoring the expression of antioxidant defense enzymes, including SOD1/2 and catalase inhibited by glucolipotoxicity. (**d**) Representative images of mitochondrial membrane potential by JC-1 staining, and cells observed with green and red fluorescence, are also quantified in percentages. The increase of JC-red shows that Liraglutide protects against glucolipotoxicity-induced mitochondrial dysfunction (JC-green). Scale bar: 50 μm. All values are presented as the mean ± SEM. Significant difference was determined using multiple comparisons of Dunnett’s posthoc test for * *p* < 0.05 and ** *p* < 0.01.

**Figure 3 biomolecules-12-01338-f003:**
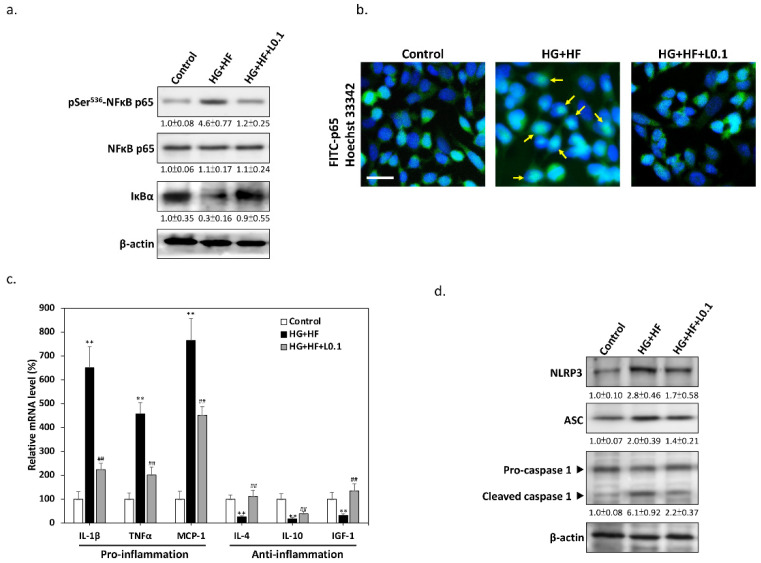
Liraglutide inhibits glucolipotoxicity-induced NFκB pro-inflammatory responses. (**a**) Western blots showed that 0.1 μM Liraglutide inhibits glucolipotoxicity-induced NFκB p65 subunit phosphorylation at Ser^536^ and prevents IκBα degradation at 24 h. (**b**) Double fluorescence staining revealed that Liraglutide inhibits the entry of the NFκB p65 subunit into the nucleus under glucolipotoxicity. Scale bar: 20 μm. (**c**) qPCR results demonstrated that under glucolipotoxicity, liraglutide significantly reduces the mRNA expression levels of the IL-1β, TNFα, and MCP-1 pro-inflammatory cytokines, and simultaneously restores the IL-4, IL-10, and IGF-1 anti-inflammatory cytokines. (**d**) Western blot analysis of NLRP3, ASC, and cleaved-caspase-1 confirmed that Liraglutide suppresses glucolipotoxicity-induced inflammasome activation. Arrows indicate nuclear p65 immunolabeling. All values are presented as the mean ± SEM. Significant difference was determined using multiple comparisons of Dunnett’s posthoc test for ** *p* < 0.01 compared to the Control group; ## *p* < 0.01 compared to the HG+HF group.

**Figure 4 biomolecules-12-01338-f004:**
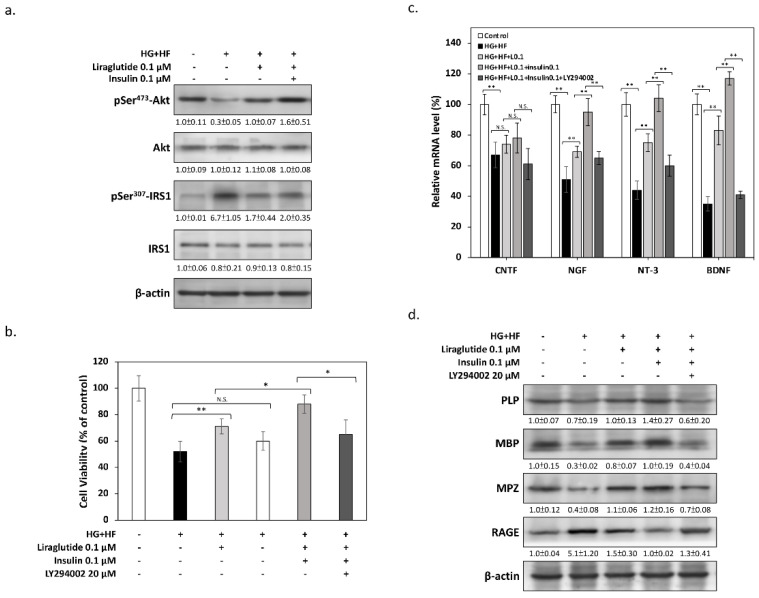
Liraglutide enhances the normal physiological function of SCs via insulin-Akt signaling. (**a**) Western blot analysis of Ser^473^-phosphorylated Akt confirmed that Liraglutide and insulin reverse the glucolipotoxicity-induced insulin signaling blockade. (**b**) MTT assays showed that the protective effect of Liraglutide and insulin were inhibited by co-treatment with 20 μM LY294002. (**c**) mRNA levels of neurotrophic factors, including CNTF, NGF, NT-3, and BDNF, were measured by qPCR. Liraglutide and insulin significantly elevated the mRNA levels of neurotrophic factors suppressed by glucolipotoxicity. However, LY294002 counteracted the effects of Liraglutide and insulin. (**d**) Western blots demonstrated that Liraglutide and insulin show efficacy in improving SC synthesis of essential myelin components and decrease the expression of the demyelination marker RAGE. Similarly, LY294002 blocked the effects of Liraglutide and insulin in promoting myelination in RSC96 SCs. All values are presented as the mean ± SEM. Significant difference was determined using multiple comparisons of Dunnett’s posthoc test for * *p* < 0.05 and ** *p* < 0.01. N.S., not significant.

**Table 1 biomolecules-12-01338-t001:** Primer sequences of different genes for qPCR analysis.

Genes	Forward	Reverse
IL-1β	5′-CACCT CTCAA GCAGA GCACA G-3′	5′-GGGTT CCATG GTGAA GTCAA C-3′
TNFα	5′-AAATG GGCTC CCTCT CATCA GTTC-3′	5′-TCTGC TTGGT GGTTT GCTAC GAC-3′
MCP-1	5′-GTGCT GACCC CAATA AGGAA -3′	5′-TGAGG TGGTT GTGGA AAAGA-3′
IL-4	5′-TGCAC CGAGA TGTTT GTACC-3′	5′-GGATG CTTTT TAGGC TTTCC-3′
IL-10	5′-GCAGG ACTTT AAGGG TTACT TGG-3′	5′-GGGGA GAAAT CGATG ACAGC-3′
IGF-1	5′-TGCTT CCGGA GCTGT GATCT-3′	5′- CGGAC AGAGC GAGCT GACTT-3′
CNTF	5′-ATGGC TTTCG CAGAG CAAAC-3′	5′-CAACG ATCAG TGCTT GCCAC-3′
NGF	5′- AGGCT TTGCC AAGGA CG-3′	5′-CCAGT GGGCT TCAGG GA-3′
NT-3	5′-CGTGG TGGCG AACAG AACAT-3′	5′-GGCCG ATGAC TTGTC GGTC-3′
BDNF	5′-GCTGG CGATT CATAA GGATA GAC-3′	5′-TATAC AACAT AAATC CACTA TCTTC CCCT-3′
GAPDH	5′-TGGTAT CGTGG AAGGA CTCAT GAC-3′	5′-ATGCC AGTGA GCTTC CCGTT CAGC-3′

## Data Availability

The data that support the findings of this study are available from the corresponding author, C.-L.L., upon reasonable request.

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
