# Peer review of "Liraglutide Attenuates Glucolipotoxicity-Induced RSC96 Schwann Cells’ Inflammation and Dysfunction"

_biomolecules, 2022, doi:10.3390/biom12101338_

Round 1

Reviewer 1 Report

This subject is of great importance given the significant effect of liraglutide on Schwann cells inflammation and dysfunction can cause and the little known regarding peripheral diabetic neuropathy. However, I would suggest major revisions for this study to be perfect. 

In general,

1. Images of all immunocytochemistry showed low resolution and quality. Authors should change all images using high-resolution images and indicate significant signals using arrows

2. There were no quantitative graphs for western blot and ICC images. The quantitative analysis should be needed.

3. With only in vitro evidence, concluding that liraflutide effectively attenuates glucolipipotoxicity and then affects the pathophysiology of DPN may be not appropriate. So, authors should show ex vivo or in vivo evidence, not necessarily what you’d use transgenic animal models.

Specifically,

4. Figure 1: If GLP1-R is expressed in Schwann cells, authors should show its location in in vivo peripheral nerves, for example, nerve injury model and so on.

5. Figure 2: In Figure 2c, the authors should show expression patterns of well-known prooxidants proteins in Schwann cells.

6. Figure 3 and 4: Authors should show the relationship between NF-kB and myelin-related genes, for example using IP between NF-kB and P0.

Author Response

Reviewer #1:

This subject is of great importance given the significant effect of liraglutide on Schwann cells inflammation and dysfunction can cause and the little known regarding peripheral diabetic neuropathy. However, I would suggest major revisions for this study to be perfect.

In general,

  1. Images of all immunocytochemistry showed low resolution and quality. Authors should change all images using high-resolution images and indicate significant signals using arrows

Ans: Many thanks to Reviewer for the valuable suggestions. We have provided higher resolution images in this version, and added arrows in Fig. 3b by your instructions, hopefully this change will make our results clearer.

  1. There were no quantitative graphs for western blot and ICC images. The quantitative analysis should be needed.

Ans: As suggested, we have added quantitative results to this revision, including all WB bands and the images of JC-1 staining.

  1. With only in vitro evidence, concluding that liraflutide effectively attenuates glucolipipotoxicity and then affects the pathophysiology of DPN may be not appropriate. So, authors should show ex vivo or in vivo evidence, not necessarily what you’d use transgenic animal models.

Ans: Because so far, no animal model can fully and accurately reflect the real progression and symptoms of human diabetic neuropathy. Moreover, most of the relevant animal experiments at present are focused on exploring the feeling of pain, and the research on glial cells is still very rare. Since our study focused on Schwann cells, and the limited time available to respond, it is unlikely that animal experiments could be performed in this revised version. However, researchers have previously conducted experiments with streptozotocin-diabetic rats and found that administration of GLP-1R agonist exenatide for 8 weeks did improve neurological symptoms (Diabetes Obes Metab. 2011; 13: 990–1000). However, the role of SCs cannot be confirmed due to the difficulty in directly separating neurons and glial cells for analysis. In our study, we simulated the most common HFD/STZ-diabetic model in the form of high FFAs+glucose, and investigated the GLP-1 signaling of SCs. We believe that the results found in this study can serve as the basis for our future animal experiments to establish a more suitable model of diabetic neuropathy, so as to further explore and verify whether regulating GLP-1 signaling in SCs is a worthy anti-DN strategy. Anyway, thanks again for your valuable suggestions.

Specifically,

  1. Figure 1: If GLP1-R is expressed in Schwann cells, authors should show its location in in vivo peripheral nerves, for example, nerve injury model and so on.

Ans: Due to the limited response time, and the fact that animal experiments on diabetic neuropathy usually take a long time to induce, it is obviously impossible for us to provide in vivo data in this revision. In fact, not only our results in Fig. 1b, but many other studies have also confirmed that SCs can indeed express GLP-1R, plus administration of GLP-1 RA also affects the physiological functions of SCs such as survival or myelination (Int J Mol Sci. 2021; 22: 2971), so we believe it should be very plausible that SCs can express GLP-1R in vivo.

  1. Figure 2: In Figure 2c, the authors should show expression patterns of well-known prooxidants proteins in Schwann cells.

Ans: As suggested, we have added WB results of 4-HNE, a biomarker of oxidative stress-related protein adducts in Fig. 3b.

  1. Figure 3 and 4: Authors should show the relationship between NF-kB and myelin-related genes, for example using IP between NF-kB and P0.

Ans: In fact, it is already a known fact that NFκB promotes demyelination of SCs. Especially in the pathogenic process such as multiple sclerosis, NFκB signaling has been confirmed to be the main cause of inflammatory demyelinating, that is, NFκB activation indeed can inhibit the expression of myelin-related proteins (Front Immunol. 2020;11:391). However, since p50/p65 usually acts as a transcriptional factor to affect the promoter of the target gene to change the expression of the protein (such as myelin-related proteins), we believe that direct co-IP between p50/p65 and myelin-related proteins may still be difficult to answer this question raised by the reviewer. However, from our results in Fig. 4c and 4d, it can be observed that both neurotrophic factors and myelin-related proteins are significantly inhibited after treatment with LY294002. Considering that LY294002 is not only an inhibitor of PI3K, but also an inhibitor of NFκB entry into the nucleus (Biochem Pharmacol. 2012;83:106-14), we think this result seems to be more able to explain the effect of NFκB signaling on the expression of myelin-related proteins in SCs. It is expected that in the future we will design additional experiments to directly inhibit NFκB signaling by means of mRNA knockdown to further clarify its exact role in the myelination of SCs.

Reviewer 2 Report

The authors reported the protective activities of liraglutide toward RSC96 Schwann cells (SCs) exposed to a high-glucose (30 mM) and high FFAs (250 mM) condition. The topic is interesting, and the findings may be relevant to the pathogenesis of diabetic neuropathy. However, several issues need to be addressed.

1.     The concentration of FFAs (250 mM) seems super-physiological. The authors need to verify that the SC death is due to glucolipotoxicity, but not hypertonicity. This reviewer recommends the authors to use 250 mM of Mannitol, instead of FFAs.

2.     The author stated that the structural and functional abnormalities of the sensory neurons are related to the pathogenesis of DN. Therefore, it is desirable to examine the effects of glucolipotoxicity on cultured neurons, in addition to SCs.

3.     It is unclear whether the glucolipotoxicity-induced SC death and other abnormalities are general phenomena or RSC96-specific. This reviewer recommends the authors to verify the alleviating effects of Liraglutide by using primary cultured SCs and/or other lined SCs.

4.     This reviewer recommends the authors to examine the downstream molecules of insulin signaling, such as IRS and PDK. These analyses would make the authors’ hypothesis (Lines 374-378) more convincing.

5.     The manuscript is basically comprehensible, but there are a plenty of grammatical errors (Line 22, 49, 60, 79, 84, 226, etc.). The authors need to ask a native English speaker to revise the sentences.

6.     Minor points:

1)    Page 1; Abstract should be more concise based on the findings.

2)    Page 2-3; the primary antibodies used in Western blot analysis and immunofluorescence assay should be indicated as a table (including a dilution for use). All the materials used in the study (including the inhibitors) should be depicted.

3)    Pages 3-4; the secondary antibodies used in Western blot analysis and immunofluorescence assay should be indicated (including a dilution for use).

4)    Page 4; The sequences of the PCR primers should be indicated as a table.

5)    Page 6; it seems over-speculative that the authors indicate signaling pathways based on the findings of Western blotting. Some quantitative analyses in Fig.2 are desirable.

6)    Fig.3b; the fluorescence pictures are of poor quality.

7)    Fig.4b; it is desirable to show the data for [HG+HF+insulin].

8)    Page 10, Lines 369-370; the authors observed mRNA expression, but not secretion, of neurotrophic factors. More careful description is required.

Author Response

Reviewer #2:

The authors reported the protective activities of liraglutide toward RSC96 Schwann cells (SCs) exposed to a high-glucose (30 mM) and high FFAs (250 mM) condition. The topic is interesting, and the findings may be relevant to the pathogenesis of diabetic neuropathy. However, several issues need to be addressed.

  1. The concentration of FFAs (250 mM) seems super-physiological. The authors need to verify that the SC death is due to glucolipotoxicity, but not hypertonicity. This reviewer recommends the authors to use 250 mM of Mannitol, instead of FFAs.

Ans: Since we do not know whether mannitol has unexpected effects on SCs, we used another commonly used sucrose to evaluate osmolarity and is equally safe as an alternative. As suggested, we have added 250 mM of sucrose instead of FFAs in Fig. 1a. The result showed that the cytotoxicity of SCs is indeed mainly induced by FFAs, not by the hypertonic environment caused by sucrose.

  1. The author stated that the structural and functional abnormalities of the sensory neurons are related to the pathogenesis of DN. Therefore, it is desirable to examine the effects of glucolipotoxicity on cultured neurons, in addition to SCs.

Ans: In fact, in our other studies, FFAs also caused significant cytotoxicity to the peripheral neuronal cell line PC12 under the same condition. Similarly, it has been shown earlier that palmitic and stearic fatty acids can indeed induce significant cell death in differentiated PC12 cells (J Neurochem. 2003;84:655-68). Although the interaction between SCs and neuron indeed contributes to the progression of DN, since this study mainly focuses on the response of SCs under glucolipotoxicity, we hope not to present its effect on neuron for the time being, and avoid presenting already the already existing known and repeatable research results.

  1. It is unclear whether the glucolipotoxicity-induced SC death and other abnormalities are general phenomena or RSC96-specific. This reviewer recommends the authors to verify the alleviating effects of Liraglutide by using primary cultured SCs and/or other lined SCs.

Ans: As far as we know, isolation and culture of Schwann cells is a complex and difficult process. Due to the limited time available, we believe that re-running the SC primary culture to test the inferences of this study may be difficult. We will carefully add the reviewer's high-value suggestions to the research targets to be carried out in the future, and also ask the reviewer to understand that it is impossible to complete according to the current situation.

  1. This reviewer recommends the authors to examine the downstream molecules of insulin signaling, such as IRS and PDK. These analyses would make the authors’ hypothesis (Lines 374-378) more convincing.

Ans: As suggested, we have added pIRS-1/IRS-1 in Fig. 4a. It can be seen from the results that Liraglutide can indeed restore IRS-1 serine phosphorylation (marker of insulin signaling blockade) induced by glucolipotoxicity.

  1. The manuscript is basically comprehensible, but there are a plenty of grammatical errors (Line 22, 49, 60, 79, 84, 226, etc.). The authors need to ask a native English speaker to revise the sentences.

Ans: Thank you for showing these grammatical or rhetorical errors. In addition to the wrong sentences you mentioned, we have also asked our native English speaker colleagues to help us correct the entire manuscript. I hope the revisions in this version can improve the accuracy and readability.

  1. Minor points:

1) Page 1; Abstract should be more concise based on the findings.

Ans: As suggested, the abstract has been rewritten.

2) Page 2-3; the primary antibodies used in Western blot analysis and immunofluorescence assay should be indicated as a table (including a dilution for use). All the materials used in the study (including the inhibitors) should be depicted.

Ans: The primer sequence list has been amended in Tab. 1 as requested. In addition, all Cat# No. of antibodies were shown in this revision. The dilution ratio is also annotated in materials and methods.

3) Pages 3-4; the secondary antibodies used in Western blot analysis and immunofluorescence assay should be indicated (including a dilution for use).

Ans: This part has been amended as requested. The secondary antibodies were used at 1:1000-1:5000 dilutions (ICC and WB), depending on individual samples condition.

4) Page 4; The sequences of the PCR primers should be indicated as a table.

Ans: The sequences of the PCR primers have been rearranged into Tab. 1.

5) Page 6; it seems over-speculative that the authors indicate signaling pathways based on the findings of Western blotting. Some quantitative analyses in Fig.2 are desirable.

Ans: As requested, we have added quantitative results to this revision, including all WB bands and the images of JC-1 staining.

6) Fig.3b; the fluorescence pictures are of poor quality.

Ans: We have added higher resolution fluorescence images in this revision.

7) Fig.4b; it is desirable to show the data for [HG+HF+insulin].

Ans: The data for HG+HF+insulin has been added in this figure.

8) Page 10, Lines 369-370; the authors observed mRNA expression, but not secretion, of neurotrophic factors. More careful description is required.

Ans: The narrative for this paragraph has been rewritten. All sentences referring to “secretion” have been corrected to prevent errors of over-inference. And thanks again to reviewer for all the corrections to our work.

Round 2

Reviewer 1 Report

The revised manuscript fulfills the requirements for publication in Antioxidants.

Reviewer 2 Report

No further comments.